# Effects of Smart Glasses on the Visual Acuity and Eye Strain of Employees in Logistics and Picking: A Six-Month Observational Study

**DOI:** 10.3390/s24206515

**Published:** 2024-10-10

**Authors:** Robert Herold, Hayarpi Gevorgyan, Lukas S. Damerau, Ulrich Hartmann, Daniel Friemert, Rolf Ellegast, Christoph Schiefer, Kiros Karamanidis, Volker Harth, Claudia Terschüren

**Affiliations:** 1Institute for Occupational and Maritime Medicine (ZfAM), University Medical Center Hamburg-Eppendorf (UKE), 20459 Hamburg, Germany; hayarpi.gevorgyan@ias-gruppe.de (H.G.); l.damerau@uke.de (L.S.D.); harth@uke.de (V.H.); claudia.terschueren@haw-hamburg.de (C.T.); 2Department of Mathematics and Technology, University of Applied Sciences Koblenz, 53424 Remagen, Germany; hartmann@hs-koblenz.de (U.H.); friemert@hs-koblenz.de (D.F.); 3Institute for Occupational Safety and Health of the German Social Accident Insurance (IFA), 53757 Sankt Augustin, Germany; rolf.ellegast@dguv.de (R.E.); christoph.schiefer@dguv.de (C.S.); 4Sport and Exercise Science Research Centre, School of Applied Sciences, London South Bank University, London SE1 0AA, UK; k.karamanidis@lsbu.ac.uk; 5Department of Sport Science, Faculty for Mathematics and Natural Sciences, University of Koblenz, 56070 Koblenz, Germany

**Keywords:** smart glasses, visual acuity, eyesight, eye strain, logistics, picking, field study, longitudinal study

## Abstract

The usage of smart glasses in goods logistics and order picking has mainly been studied through cross-sectional experimental studies. Our longitudinal field study investigated the effects of smart glasses on the eyesight of 43 employees at two German companies. We combined ophthalmological examinations and questionnaire surveys at two points in time, six months apart. The vision of the employees was examined before and after each work shift. Mixed effects logistic regression was conducted to determine the associations between smart glasses use and effects on visual acuity. In the baseline examination, differences in eyesight before and after shifts were small and not statistically significant. However, some individuals experienced deteriorations, especially in visual acuity at near distances (*n* = 7 for the right eye, *n* = 6 for the left). Participants over 40 years of age had 16.1 times higher odds of deterioration compared to those under 40 years (95% CI: 2.7–95.9, *p* = 0.002). The most commonly reported eye strains were eye fatigue (*n* = 32), rubbing (*n* = 25), and burning (*n* = 24). If smart glasses are to be implemented in logistics companies, it is recommended to offer employees eye tests with an industrial physician in advance.

## 1. Introduction

As part of the ongoing digitalization of the German economy, companies are testing monocular and binocular smart glasses, i.e., in goods logistics and order picking, instead of handheld scanners in the work process. The use of monocular or binocular smart glasses at workplaces for several hours is not only a technical innovation, but it also introduces a new way of displaying task-relevant information close to the eye [1]. A number of feasibility studies have been conducted on the use of smart glasses in industrial workplaces [2]. However, because smart glasses have not been used in everyday work for long—only a few employees have worked with smart glasses in pilot testing—any resulting health effects have been, accordingly, little investigated. Therefore, the health consequences of the long-term use of smart glasses at workplaces, such as for the eyes, have not yet been assessed.

In studies examining the possible health consequences of wearable eye technologies, such as virtual reality (VR) or augmented reality (AR) glasses, or so-called head-mounted displays (HMDs), users report headaches, aching eyes, and fatigue. Another finding from these studies is that users report having less fun with augmented reality when they associate the devices with health risks, indicating possible lower adoption of the innovative technology due to health concerns [3]. Additionally, in another study on eye health and monocular optically transparent AR displays, Gabbard et al. (2018) reported visual fatigue for near vision as well as distance vision. However, participants experienced more fatigue when working with a head-mounted display with a real screen compared to when working with a physical screen [4].

Sheppard and Wolffsohn (2018) described so-called “sicca syndrome” as a possible effect of smart glasses, where users develop dry eyes [5]. The fact that the daily use of smart glasses at workplaces for several hours could have health effects can also be inferred from available studies on computer users. Since working with smart glasses is similar to working with a computer screen, and the display is even placed much closer to the eye, the same symptoms could also occur. Messmer et al. (2015) reported frequent headaches and dry eyes in computer users [6].

Bogdănici et al. (2017) described a reduction in accommodative or binocular vision ability and computer vision syndrome (CVS), which includes symptoms such as burning, twitching, and tired eyes, blurred or double vision, headaches, sensitivity to light, and shoulder and neck pain [7]. Hwang et al. (2021) conducted research on CVS and concluded that approximately 70% of individuals who frequently use screens have CVS [8]. Sánchez-Valerio et al. (2020) observed in their study of screen activity on stationary personal computers (PCs) that participants experienced dryness in their eyes. A total of 108 subjects, aged 18 to 45 years with no pre-existing conditions, were studied in three groups. The first group worked at a computer screen for less than 4 h per day, the second group for 4 to 7.9 h per day, and the third group for more than 8 h. The results showed a significant correlation between the duration of screen use and dryness in the eyes of participants [9]. Choi et al. (2018) described that smartphone use, i.e., the frequent reading of information on a typically small display, leads to subjective complaints such as eye fatigue or burning of the eyes, which are also labeled as digital eye strain (DES) [10].

In a review of the literature, Friemert et al. (2020) stated that most results on eye strain from smart glasses are based on experiments with a small number of young subjects [11]. These studies examined context and focal distance switching [4], level of sickness symptoms (SSQs) [12], visual fatigue [13], reaction time [14], and sensitivity of the visual field [15]. Currently, there is a need for research on possible health effects specifically related to the use of smart glasses in workplaces.

To contribute to this research, the observational study “*Auswirkungen von Datenbrillen auf Arbeitssicherheit und Gesundheit*” [Effects of Smart Glasses on Occupational Safety and Health] (ADAG) was conducted in two companies in the logistics industry and in the quality control of an assembly unit. The study aimed to quantify the effects of monocular smart glasses on the vision and eye health of employees working in goods logistics and picking.

## 2. Materials and Methods

An observational field study with two time points was conducted in two different companies across two sites in Germany. The employees in both companies used Google Glass Enterprise Edition smart glasses (Figure 1) from Google LLC (Mountain View, CA, USA), which were no longer being sold at the time of writing [16]. The smart glasses had an optical display attached on the upper-right side of the glasses frame. The employees took part during their normal working hours. All examinations were carried out on site at the companies. As planned, a medical eye examination was conducted on the same day before and after a work shift during the fieldwork phase. The subjective parameters of the eye strain experienced by participants while using the smart glasses in their everyday work were recorded with a self-completion questionnaire, which included, among other things, questions on age, visual aid, and sleep. The questionnaire was provided in German and was written in simple language. Whenever possible, pictures were included for better understanding. The investigators on-site were available to assist with questions in German, English, and Russian.

Diagnoses of exclusion were adopted analogously to the study on eye health on occupational computer users by Segui et al. (2015): corneal leukoma, amblyopia, tropia, myopia surgery, keratoconus, inflamed pinguecula, allergic conjunctivitis, blepharitis, cataract surgery, anisometropia, chronic keratitis, traumatic cataract, pseudophakia, pterygium, diabetic retinopathy, glaucoma, myopia magna, and uveitis [17].

The study protocol was approved by the ethics review board of the Koblenz University of Applied Science. We hypothesized that wearing smart glasses in the workplace would have various effects on eye health and user acceptance. The examination of eye health was planned at two time points, while acceptance was to be assessed at three time points, each across two companies. A 2 × 3 repeated measures ANOVA indicated a required sample size of *n* = 58, assuming an alpha of 0.05, a power (β) of 0.80, and a small to medium effect size (f = 0.16).

In the first fieldwork phase (T0), a total of 43 participants were recruited from two companies, with 32 participants from company 1 (August 2019) and 11 participants from company 2 (April 2019). In the second fieldwork phase (T1), ten participants from company 1 and seven participants from company 2 were re-examined after six months. The T1 examination was conducted in February 2020 for company 1 and in October 2019 for company 2. For the longitudinal comparison, the measurement results of the eye examination before the shift were available for a total of 17 subjects after 6 months.

In this study, an Optovist EU vision screening device from Vistec AG (Olching, Germany) [18] was used for the eye examinations. All eye examinations were carried out by an occupational physician. Landolt rings were used to examine visual acuity at a far distance of 6 m and at a near distance of 0.40 m for both the left and right eyes. Visual acuity was measured in steps, with the vision screening device capable of measuring within a range from 0.10 (worst value) to 1.25 (best value). Due to time constraints during the examinations, visual acuity measurements were conducted solely within the range of 0.25 to 1.00. Further examination parameters were a phoria test (binocular), a stereopsis test (binocular), a color perception test (binocular), a glare test (right eye, left eye), and an Amsler grid test (right eye, left eye). The Amsler grid test, used for the early detection of retinal diseases such as macular degeneration, was conducted to identify any existing, previously undetected defects in the central visual field or distorted image perception, and to determine whether these could be triggered by wearing the monocular smart glasses for several hours.

We analyzed the questionnaire data by using descriptive statistics such as the mean (MV), standard deviation (SD), minimum (min), and maximum (max) for metric variables. For ordinal variables, we presented absolute and relative frequencies and indicated missing values wherever applicable. The results were presented separately for each company. Additionally, we used descriptive statistics including the MV, SD, min, and max to present the results of the eye examination. Since all of the participants were able to complete their work tasks, the minimum visual acuity for descriptive analyses was set at 0.10 if a level of 0.25 was not achieved. Given the low number of participants, we primarily used non-parametric tests. To evaluate changes in visual acuity over time (before and after the shift), we used Wilcoxon signed-rank tests, also known as paired samples Wilcoxon tests.

Generalized linear mixed models, which incorporate both fixed effects parameters and random effects in a linear predictor, were fitted using maximum likelihood estimation. The Laplace approximation was used to evaluate the log-likelihood. Mixed effects logistic regression models were used to investigate which parameters influence the deterioration in visual acuity during the shift (short-term effects) and after six months (long-term effects). For this purpose, all initial measurements lower than 0.25 were excluded. The dependent variable was defined as either the decrease in visual acuity after the shift compared to before the shift (model 1 for short-term effects) or the decrease in visual acuity at the follow-up compared to the baseline, measured before the shift in each case (model 2 for long-term effects). Visual acuity at near and far distances, as well as in the right and left eyes, was examined in a joint model. Random intercept models were used, and the model for the short-term effects was adjusted for eye side, distance, gender, age, sleep quality, and the requirement of a visual aid. Due to the lower number of participants in the follow-up, no adjustment was made for sleep quality and visual aid in the model for the long-term effects.

A logistic regression model was used to investigate which parameters influence reported eye strain at the baseline examination. Only participants who answered all five questions on eye strain (glare, afterimages, eye fatigue, burning, or rubbing) were included in the analysis. The dependent variable of the model was whether more than three eye strain symptoms were reported. The model was adjusted for gender, age, sleep quality, and the requirement of a visual aid.

We reported exact *p*-values and confidence intervals to provide a clearer interpretation of the statistical significance. All statistical analyses were performed using R Statistical Software (The R Foundation for Statistical Computing, Vienna, Austria), version 4.4.1 [19]. The lme4 package [20] was used to perform the mixed effects logistic regression, while the ggplot2 [21] package was utilized for graphical representation.

## 3. Results

### 3.1. Study Population

Table 1 reveals that the participants were predominantly male, with a mean age of 39.2 years (SD: 11.6) at the baseline. At the six-month follow-up, 17 out of 43 participants (39.5%) participated. The average age of the remaining participants was 38.0 (10.7) years. Company 1 had the higher proportion of women at 34.4% (*n* = 11 out of 32 participants). At the baseline, 16 participants (37.2%) were of German nationality, while 27 participants (62.8%) held other nationalities. The majority of the employees were employed as pickers (85.7% at the baseline), followed by forklift drivers (7.1%), warehouse clerks (4.8%), and warehouse logistics specialists (2.4%). On a scale of 0 to 5, with 5 being very good, health received an average score of 3.9, and sleep quality was given an average score of 3.3. The percentage of participants with corrective glasses, due to hyperopia or myopia, was 31.2% in company 1 and 27.3% in company 2. No participant reported using contact lenses. While 30.2% of all of the participants reported wearing a visual aid, only 18.6% completed the eye examination with a visual aid.

### 3.2. Medical Eye Examination

Table 2 displays the eye examination results of both companies, which were conducted before and after the work shift with the monocular smart glasses. Of the 43 participants, those who had changes in values between the pre-examination and the follow-up examinations generally showed better eyesight, including at both far and near distances. However, the difference in values before and after the shift was not statistically significant. Prior to the work shift, all participants had normal results on the binocular phoria test. After the shift, only one participant had an abnormal result. In the color perception test, one person had an abnormal score before the work shift, but this abnormality did not appear after the test. The results of the glare test and the Amsler grid test were normal for all participants. Overall, these findings are considered to be within normal limits.

#### 3.2.1. First Fieldwork Phase

While there is an improvement in ocular parameters on average, some individuals experienced a deterioration in their visual function. The most common deteriorations were in visual acuity at a near distance for the right eye (*n* = 7 out of 43 participants, 16.3%) and left eye (*n* = 6, 14.0%). Table 3 reveals that age and requiring visual aids significantly impacted the deterioration of visual acuity. Participants aged over 40 had 16.1 times the odds of visual acuity deterioration compared to those under 40 (CI 95%: 2.7–95.9, *p* = 0.002). The odds of deterioration of visual acuity of those who required a visual aid were only 0.09 times that of those without a visual aid (CI 95%: 0.01–0.67, *p* = 0.019). There were no statistically significant differences between near and far distances or between the right and left eyes.

#### 3.2.2. Second Fieldwork Phase and Comparison and Long-Term Effects

We were able to examine a total of 17 individuals during the follow-up phase. The results indicated that there were no statistically significant differences in the visual acuity measurements, either for far or near distances. In the mixed logistic regression model for long-term effects, there were no significant influencing factors for the deterioration of visual acuity. However, the effect of age (OR = 5.1, not statistically significant) exhibited the same direction as in the model for short-term effects, as illustrated in Figure 2.

### 3.3. Reported Eye Strain

Although the deterioration of ocular examination parameters was rare, a large number of participants reported eye strain. The most commonly reported complaints were eye fatigue (*n* = 32 out of 37 respondents, 86.5%), rubbing (67.6%), and burning (64.9%), as shown in Table 2. Of the 35 participants who answered all five questions, *n* = 16 (45.7%) reported more than three instances of eye strain. The logistic regression model presented in Table 4 demonstrates that gender was an influential factor for the odds of reported eye strain. The odds for women were found to be 0.02 times lower than those for men (CI 95%: 0.00–0.20, *p* = 0.004). Participants over the age of 40 tended to report eye strain more frequently than younger participants (OR = 4.2, not statistically significant), as illustrated in Figure 3.

## 4. Discussion

The results of the study indicated that the visual acuity of employees who work with data glasses generally did not change over the course of a shift and over a period of six months. Nevertheless, there were groups that had an increased chance of deterioration. Eye strain was frequently reported after working with the data glasses.

Our study pointed out that employees aged 40 years and older are at risk for deteriorations of visual acuity, which is consistent with the findings of Yeow et al. (1991), who examined computer users. They reported that subjects over 40 years of age and computer users experienced accommodation problems more frequently than occupational computer users under 40 years of age [22,23]. In this older age group, age-related limitations of eyesight, such as long-sightedness (hyperopia) or short-sightedness (myopia), may become more pronounced due to the already-slowed adaptive capacity of the eye, especially the ocular lens system.

Therefore, before introducing smart glasses in companies, employees should undergo an ophthalmological examination to clarify any undiagnosed age-related limitations of eyesight. This could preferably take place during regular examinations and consultations by occupational health physicians in the workplace. We recommend consultation and examination starting at the latest by the age of 40. These measurements should be integrated into routine workplace health promotion and occupational medical care. In doing so, the effects of work that strains the eyes can be mitigated early and potential impairments of human vision be prevented.

Furthermore, if corrective glasses are already worn for diagnosed short-sightedness or long-sightedness, employees may perform better in the eyesight test after removing the smart glasses. As published by the German *Kommission Arbeitsschutz und Normung* [Commission for Occupational Health and Safety and Standardization], prompt regulation and standardization for the use of smart glasses in the workplace are needed [24].

The differences in eyesight between the same-day pre- and post-shift examinations were very small overall in this study. This result is consistent with the findings reported by Herzog et al. (2020), who used similar eyesight tests in a laboratory setting to examine whether smart glasses could be potentially harmful to the human eye [25,26].

While the results of the baseline survey, which included 43 participants, provide a clear picture, only 17 participants took part in the follow-up. Age has a significant impact on dynamic accommodation [27]. The natural decline in accommodation usually begins around the age of 40, which may not necessarily be an effect of using smart glasses. To investigate these causal effects, future long-term studies should emphasize maintaining a high response rate at follow-up.

Eye strains, including glare, afterimages, fatigue, burning, and eye rubbing, appear to be a concern, as 45.7% of the 35 participants who answered all five questions reported experiencing more than three of the five types of eye strain inquired about. Working with AR head-mounted devices may reduce the number of blinks [28], potentially contributing to eye strain. However, in our study the number of missing answers to these five questions was higher than in the other sections of the questionnaire. Although we wrote the questionnaire in simple language, provided pictures for clarification, and offered support in German, English, and Russian, meaningful images were lacking for these five questions.

In summary, the medical examination of eyesight is an objective measurement parameter from the fieldwork phases of the ADAG study. One of the strengths of the study is that fieldwork phases were carried out in companies in logistics and picking under everyday working conditions. Using the same Optovist EU vision screening device for all participating employees avoided any bias caused by different screening devices. The tests were chosen because they can indicate effects on the eye of close exposure to a display or reflected light; they are also part of occupational health screening in Germany and can thus be carried out regularly in companies in the future. Technical, organizational, and ergonomic aspects are key challenges associated with augmented reality smart glasses in logistics and supply chain management [29]. The findings provide initial indications of the possible effects and necessary requirements for eye health when using monocular and binocular smart glasses in everyday working life. The results of this study will help to supplement existing knowledge, particularly with regard to ergonomic aspects of eye health.

The study has several limitations. Firstly, the study was conducted in a relatively young and predominantly male working population. Chowdhury et al. (2017) showed that the healthy worker effect consists of the three components of healthy hire, time since hire or survival effect, and the advantages of working [30]. It was not possible to compare to or statistically adjust for a control group from the general population. Therefore, our results are not transferable to the general population. Secondly, it must be noted that only a few employees could be recruited from the participating companies. Fewer participants (*n* = 43) than the a priori target (*n* = 58) took part in the baseline study, which could result in an inflation of the discovered associations [31]. Thirdly, only 39.5% of the initial participants took part in the second survey after six months; this is due to the frequent changing of jobs in the logistics and picking sector. Some subjects were also on holiday or unable to attend due to illness at the time of the follow-up. Overall, this low participation resulted in limited explanatory power of the longitudinal evaluation of the medical eyesight examinations. Fourthly, avoidable difficulties with the questionnaire may have resulted in 18.6% of the baseline participants not answering all of the questions about eye strain.

Observational studies in the workplace play an important role in occupational medicine [32], but they are often constrained by time and financial limitations. Our eye examinations were based on tests using Landolt rings. Willingness to participate in future studies could potentially be increased by gamifying the examinations, as suggested by a study on the impact of the gamification of vision tests on the user experience [33]. To gain a comprehensive understanding of acute and chronic effects, future research should also assess the effects of the blue light-emitting diodes (LEDs) of the smart glasses on eyesight, which could not be investigated in this study. However, this could mean possible strain on the eye’s lens and fundus of the eye if smart glasses are used for many years in everyday work. According to current knowledge, long-term effects of the blue light component of LEDs cannot be assessed at present [34]. A thermal effect on the eye due to the radiation of the information-processing data unit with a Wi-Fi connection on the side arm of the monocular smart glasses should also be considered in further studies.

## 5. Conclusions

The short-term results of our study suggest that younger participants and those who were provided with corrective glasses experienced less frequent deterioration of visual acuity. If smart glasses are to be implemented in a logistic company it is recommended to offer employees eye tests with an industrial physician in advance. To gain a better understanding of the long-term effects, further workplace studies on the use of data glasses in logistics and picking are needed.

## Figures and Tables

**Figure 1 sensors-24-06515-f001:**
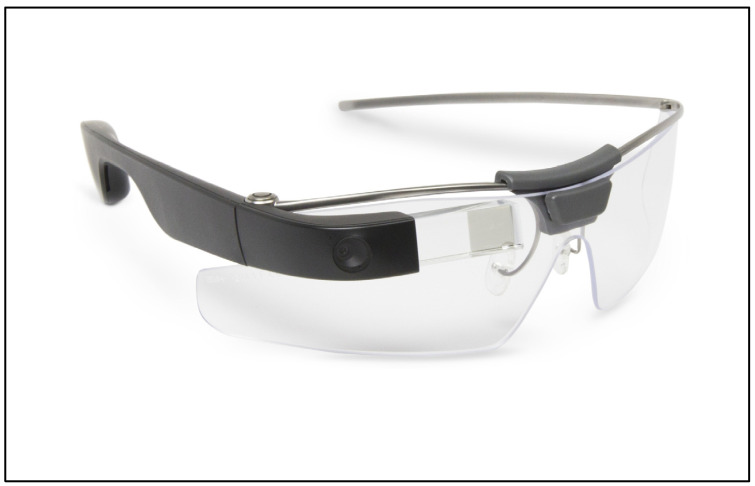
Google Glass Enterprise Edition (source: https://mashable.com/article/google-glass-enterprise-edition, accessed on 8 October 2024).

**Figure 2 sensors-24-06515-f002:**
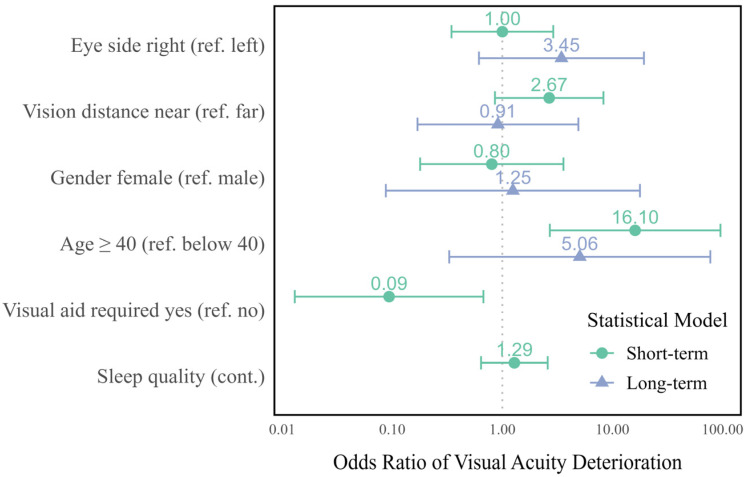
Interval plot of the odds ratios and 95% confidence intervals for deterioration in visual acuity (green: short-term effects, blue: long-term effects; visual aid and sleep quality not included in the long-term model).

**Figure 3 sensors-24-06515-f003:**
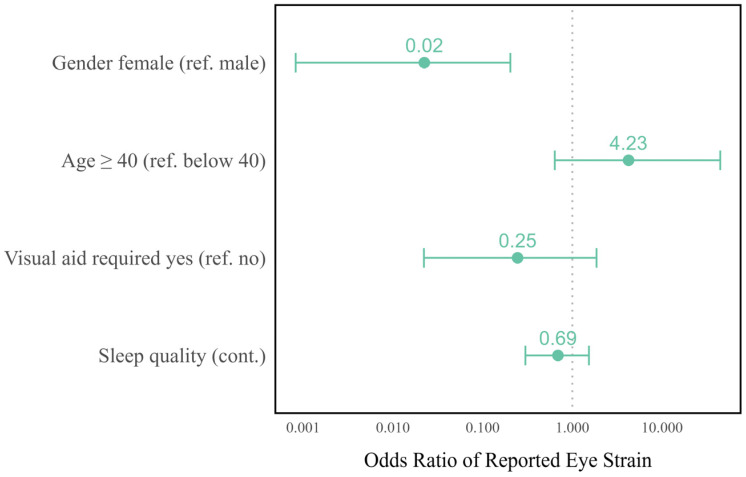
Interval plot of the odds ratios and 95% confidence intervals for reported eye strain after removing the data glasses on the day of the baseline examination.

**Table 1 sensors-24-06515-t001:** Study population.

Parameter		Baseline			Follow-Up Total (*n* = 17)
		Company 1 (*n* = 32)	Company 2 (*n* = 11)	Total (*n* = 43)	
Gender	Male	21 (65.6%)	10 (90.9%)	31 (72.1%)	10 (58.8%)
	Female	11 (34.4%)	1 (9.1%)	12 (27.9%)	7 (41.2%)
Age [years]	Mean (SD)	41.3 (11.2)	33.2 (10.9)	39.2 (11.6)	38.0 (10.7)
	Min–max	26–63	23–55	23–63	24–55
Nationality	German	12 (37.5%)	4 (36.4%)	16 (37.2%)	4 (23.5%)
	Other	20 (62.5%)	7 (63.6%)	27 (62.8%)	13 (76.5%)
Highest school degree	Elementary school	4 (12.5%)	1 (9.1%)	5 (11.6%)	2 (11.8%)
	Second. school (9 or 10 years)	15 (46.9%)	8 (72.7%)	23 (53.5%)	10 (58.8%)
	High school diploma	13 (40.6%)	2 (18.2%)	15 (34.9%)	5 (29.4%)
Vocational training	Completed apprenticeship	10 (33.3%)	6 (85.7%)	16 (43.2%)	6 (46.2%)
	University degree	6 (20.0%)	1 (14.3%)	7 (18.9%)	2 (15.4%)
	Without vocational training	14 (46.7%)	0 (0.0%)	14 (37.8%)	5 (38.5%)
	Missing	2	4	6	4
Occupation	Picker	31 (100.0%)	5 (45.5%)	36 (85.7%)	13 (81.2%)
	Other (warehouse clerk, logistics specialist, forklift driver)	0 (0.0%)	6 (54.5%)	6 (14.3%)	3 (18.8%)
	Missing	1	0	1	1
Self-rated health (very poor to very good)	Mean (SD)	3.8 (0.8)	4.4 (0.8)	3.9 (0.8)	3.7 (1.3)
	Min–max	2–5	3–5	2–5	1–5
Self-rated sleep quality in the previous night (very poor to very good)	Mean (SD)	3.1 (1.1)	3.6 (1.0)	3.3 (1.1)	-
	Min–max	0–5	2–5	0–5	-
Visual aid required	No	22 (68.8%)	8 (72.7%)	30 (69.8%)	9 (90.0%)
	Yes	10 (31.2%)	3 (27.3%)	13 (30.2%)	1 (10.0%)
	Missing	-	-	-	7
Visual aid worn during the test	No	25 (78.1%)	10 (90.9%)	35 (81.4%)	16 (94.1%)
	Yes	7 (21.9%)	1 (9.1%)	8 (18.6%)	1 (5.9%)

SD: standard deviation; min: minimum; max: maximum; and *n*: sample.

**Table 2 sensors-24-06515-t002:** Results of the medical eye examination and reported eye strain.

Parameter			Baseline (*n* = 43)		Follow-Up (*n* = 17)	
			Mean (SD)	*p*-Value *	Mean (SD)	*p*-Value *
Visual acuity far distance (values between 0.10 and 1.00)	Right eye	Pre-shift	0.87 (0.18)	0.128	0.87 (0.15)	0.672
		Post-shift	0.90 (0.16)		0.89 (0.15)	
	Left eye	Pre-shift	0.84 (0.23)	0.132	0.89 (0.22)	1.000
		Post-shift	0.87 (0.23)		0.88 (0.22)	
Visual acuity near distance (values between 0.10 and 1.00)	Right eye	Pre-shift	0.70 (0.33)	0.311	0.67(0.37)	0.584
		Post-shift	0.72 (0.34)		0.64 (0.36)	
	Left eye	Pre-shift	0.70 (0.32)	0.214	0.66 (0.36)	0.400
		Post-shift	0.73 (0.32)		0.64 (0.37)	
**Parameter**			**Baseline (*n* = 43)**			
			**Yes**		**No**	**Missing**
Glare		Post-shift	18 (50.0%)		18 (50.0%)	7
Afterimages		Post-shift	21 (60.0%)		14 (40.0%)	8
Eye fatigue		Post-shift	32 (86.5%)		5 (13.5%)	6
Burning		Post-shift	24 (64.9%)		13 (35.1%)	6
Rubbing		Post-shift	25 (67.6%)		12 (32.4%)	6

SD: standard deviation, * Wilcoxon signed-rank test with continuity correction.

**Table 3 sensors-24-06515-t003:** Deterioration of visual acuity (mixed effects logistic regression).

Parameter	Reference	Short Term (*n* = 43)				Long Term (*n* = 17)			
		Odds Ratio	95% CI		*p*-Value	Odds Ratio	95% CI		*p*-Value
			Lower	Upper			Lower	Upper	
Eye side	Left	1.00	0.35	2.91	0.998	3.45	0.61	19.40	0.160
Vision distance	Far	2.67	0.86	8.30	0.090	0.91	0.17	4.91	0.915
Gender	Male	0.80	0.18	3.59	0.774	1.25	0.09	17.80	0.870
Age	Below 40	16.10	2.70	95.90	0.002	5.06	0.33	77.60	0.245
Visual aid required	No	0.09	0.01	0.67	0.019	-	-	-	-
Sleep quality	(cont.)	1.29	0.64	2.59	0.478	-	-	-	-
Intercept	-	0.01	0.00	0.25	0.005	0.03	0.00	0.42	0.009

CI: confidence interval.

**Table 4 sensors-24-06515-t004:** Risk of reported eye strain after wearing the data glasses at the baseline (logistic regression, *n* = 35).

Parameter	Reference	Odds Ratio	95% CI		*p*-Value
			Lower	Upper	
Gender	Male	0.02	0.00	0.20	0.004
Age	Below 40	4.23	0.64	44.50	0.166
Visual aid required	No	0.25	0.02	1.86	0.198
Sleep quality	(cont.)	0.69	0.30	1.53	0.364
Intercept	-	6.74	0.41	184.65	0.210

CI: confidence interval.

## Data Availability

The data presented in this study are available on request from the corresponding author.

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
