# Peer review of "Effects of Smart Glasses on the Visual Acuity and Eye Strain of Employees in Logistics and Picking: A Six-Month Observational Study"

_sensors, 2024, doi:10.3390/s24206515_

Round 1

Reviewer 1 Report

Comments and Suggestions for Authors

Observations:

1- Join the paragraphs between lines 37 and 47 of the document.

2-Change the tense of the expression "cannot yet be assessed," found on line 48. In the paragraph, the present perfect was always used.

3-On line 88, the expression "The study aims" appears in the present tense. The project has already been executed; it should appear in the past tense.

4-On line 107, a sample size of 43 appears. How was the sample size calculated a priori? It is recommended that you express the sample value a priori and the power used for the calculation.

5-In line 113, it was stated that at the end of the six months, only 17 subjects out of 43 were evaluated. The discussion and conclusions should explain how this affects the interpretation and meaning of the data.

6-In the methodology, add an image of the smart glasses used. If possible, complement with pictures of the operators using the equipment.

7- On line 118, Why is 0.10 assumed if 0.25 is not achieved?

8-On line 130, Clarify which parameters were evaluated in the logistic regression model. Additionally, the general logistic regression model could be added, and a brief explanation could be given. The above would strengthen the methodology used.

9-Why didn't all participants answer the question posed in line 144? The impact of this restriction should be clarified in the discussion and conclusions.

10-It is recommended that the results be accompanied by graphs that complement the information and help interpret the results.

11-The discussion paragraph located between lines 245 and 251 should go in the methodology.

12-If the sentences presented in the discussion in lines 257 and 266 are statements by the researchers, why do they carry quotes 24 and 25? Please clarify and, if necessary, correct.

13-If the research focused on something other than blue light and the thermal effect of smart glasses, then it should not be speculated and confronted with the findings of other researchers. Delete the discussion paragraph located between lines 273 and 283.

14-It is recommended that the discussion be improved, strengthened, speculated, and given more meaning to the results. Indicate the effects of starting with 43 participants and moving to 17. Explain the meaning of what was obtained with the logistic regression. Add more references to the document, such as protocol support, reference the materials (smart glasses), and associate other research.

15-Adjust conclusions based on the results obtained.

16-Future work associated with the experiment and the results obtained should be added.

Comments on the Quality of English Language

The manuscript contained a few errors. The wording allows the document to be read lightly, continuously, precisely, and directly. However, it is recommended that the authors adjust the suggested observations and carry out a final check to guarantee an excellent presentation of the narrative.

Reviewer 2 Report

Comments and Suggestions for Authors

The English is clear and accurate. The aim of the study is well-defined, the methods are thoroughly conducted and explained, and both the discussion and conclusion are adequate. 

Regarding the results, "Younger participants and those who were provided with corrective glasses experienced less frequent deterioration of visual acuity" primarily refers to near visual acuity outcomes in the short term, as these were the only statistically significant results. I suggest expanding the discussion to include the natural decline in accommodation that typically begins around the age of 40-42, which may not necessarily be an effect of using these specific smart glasses.
